# An Improved Adaptive Genetic Algorithm for Two-Dimensional Rectangular Packing Problem

**Yi-Bo Li [1], Hong-Bao Sang [1],[*], Xiang Xiong [1] and Yu-Rou Li [2]**

[1] State Key Laboratory of Precision Measuring Technology and Instruments, Tianjin University, Tianjin 300072, China; slyb@tju.edu.cn (Y.-B.L.); bishop00929@tju.edu.cn (X.X.)
[2] Canterbury School, 101 Aspetuck Ave, New Milford, CT 06776, USA; yli21@cbury.org
[*] Correspondence: gs08@tju.edu.cn

**Abstract:** This paper proposes the hybrid adaptive genetic algorithm (HAGA) as an improved method for solving the NP-hard two-dimensional rectangular packing problem to maximize the filling rate of a rectangular sheet. The packing sequence and rotation state are encoded in a two-stage approach, and the initial population is constructed from random generation by a combination of sorting rules. After using the sort-based method as an improved selection operator for the hybrid adaptive genetic algorithm, the crossover probability and mutation probability are adjusted adaptively according to the joint action of individual fitness from the local perspective and the global perspective of population evolution. The approach not only can obtain differential performance for individuals but also deals with the impact of dynamic changes on population evolution to quickly find a further improved solution. The heuristic placement algorithm decodes the rectangular packing sequence and addresses the two-dimensional rectangular packing problem through continuous iterative optimization. The computational results of a wide range of benchmark instances from zero-waste to non-zero-waste problems show that the HAGA outperforms those of two adaptive genetic algorithms from the related literature. Compared with some recent algorithms, this algorithm, which can be increased by up to 1.6604% for the average filling rate, has great significance for improving the quality of work in fields such as packing and cutting.

**Keywords:** rectangular packing problem; optimization; hybrid adaptive genetic algorithm; heuristic; filling rate

## 1. Introduction

The two-dimensional rectangular packing (2DRP) problem is often involved in the manufacturing process of furniture, glass, metal, paper products, VLSI chip design, newspapers paging, and so on [1,2]. Consequently, optimized algorithms designed to improve the quality and efficiency of packing (therefore reducing the cost of the enterprises) are the subject of many researchers' explorations in recent years. The 2DRP problem can be described simply as this: rectangles of different sizes are put into a large two-dimensional rectangular container (which we call the sheet) in a certain way to maximize the filling rate of the sheet. The 2DRP is a representative NP-hard problem [3] with complex constraints, which means that there is no exact method to solve the problem in polynomial time. Thus, most researchers have devoted their efforts to fast approximate heuristics and meta-heuristics to obtain practical solutions with higher qualities.

In recent decades, in order to solve the optimal or nearly optimal packing scheme of the 2DRP problem in a compromise time and space, scholars have explored and studied many methods for solving the problem. Two types of representative approaches have been used to address the 2DRP problem: the first solves the problem directly by using different heuristics; the second uses meta-heuristic algorithms, such as Simulated Annealing (SA) [4], Genetic Algorithm (GA) [5], Particle Swarm Optimization (PSO) [6], Ant Colony Optimization (ACO) [7], etc., combined with heuristic placement strategy. Recently, other

introduced algorithms include whale optimization algorithm [8], dragonfly algorithm [9], naked mole-rat algorithm [10], moth flame optimization [11], etc.

The component in the first approach is the layout representation for packing sequence. The bottom-left (BL) algorithm [12] proposed by Baker et al. is the earliest heuristic. The main idea is to start from the top right corner of the container, making each item continuously slide as far as possible downwards and then as far as possible to the left until the item is placed in a stable position. Some BL algorithms have improved heuristic variants such as bottom-left-fill (BLF) [13], deepest bottom-left-fill (DBLF) [14], improved bottom-left (IBL) [15], and bottom-left decreasing (BLD) [16]. The lowest horizontal line algorithm (LHLA) [17] proposed by Jia et al. queries the horizontal line with the lowest height when laying out the rectangle and places the rectangle on it. However, taking account of the lack of iterative optimization, in general, the packing quality of the heuristic approach is poor.

In order to improve the packing quality, the key part of the second method is how to use meta-heuristics combined with a layout representation approach to optimize the packing sequence. Hopper et al. conducted empirical research, combining SA and GA with BL and BLF approaches [18]. The best fit algorithm (BF) [19] proposed by Burke et al. describes the sheet as a set of dynamic slots, places items on the lowest available slot at each iteration step, and then combines them together. Later, Burke's team also proposed a simulated annealing enhancement to the BF algorithm [20]. He et al. suggested an efficient deterministic heuristic [21] using the action space. Wei et al. investigated an adaptive selection approach [22] without any control parameter to address the 2D rectangle packing area minimization problem. Recently, Xia et al. proposed an adaptive genetic simulated annealing algorithm [23] applied to the problem of packing optimization of rectangles.

In view of the premature problem of Simple Genetic Algorithm (SGA) in practical applications, it is easy to fall into local optimal solution prematurely as well as poor local search ability [24]. Srinivas et al. [25] proposed an Adaptive Genetic Algorithm (AGA) that dynamically adjusts the crossover probability and mutation probability according to the fitness of the individual. However, if the fitness of the individual is close to or equal to the maximum fitness of the population, the crossover probability and mutation probability will be close or equal to zero, making the evolution of genetic algorithm stagnant. Ren proposed an improved AGA [26] for this problem. However, when the average fitness is close to the maximum fitness of the contemporary population, it is easy to cause a large number of individuals to have a lower crossover probability and mutation probability, which will stagnate the evolution. This improved AGA has also been applied to many fields such as the three-dimensional container loading problem [27] and the laminate stacking sequence optimization [28]. In addition, the algorithm proposed by Ren is only in view of individual fitness. This algorithm adaptively changes the crossover probability and mutation probability from the local perspective, but the dynamic adjustment of crossover and mutation are not considered whether they have a positive impact on the population evolution process from the overall perspective of population evolution. According to the evolution of later generations, the AGA [29] proposed by Jiang recorded the number of algebras in which the fitness of the optimal individual of the population did not change and dynamically adjusted the crossover and mutation probabilities. However, the algorithm has two shortcomings: (1) the incremental value and decrement value of crossover probability and mutation probability are greatly affected by fitness, and if the value range of the fitness function is not between 0 and 1, the adaptive algorithm will no longer be applicable; (2) although the crossover probability and mutation probability of each generation are dynamically adjusted, the crossover probability and mutation probability of all individuals in a certain generation are the same, and the performance is consistent for individuals with different fitness.

In this paper, in order to overcome the shortcomings of the two kinds of adaptive genetic algorithms, an improved method called hybrid adaptive genetic algorithm (HAGA) is proposed to supplement Ren and Jiang's algorithms. It is the key for HAGA to work

through the joint action of individual fitness from local perspective and the global perspective of population evolution. The presented algorithm not only can obtain differential performance for individuals but also deals with the impact of dynamic changes on population evolution to quickly find a further improved solution. First of all, through the research and analysis of the 2DRP, a two-dimensional packing model is established to maximize the filling rate with relevant constraints. Then, the initial population is constructed by two-stage encoding, and the heuristic lowest horizontal line algorithm is used in the process of placement to decode the packing solution and calculate the individual fitness. After the termination rules of the design are satisfied in the iterative optimization process of HAGA, the packing scheme with the largest filling rate is selected in the feasible solutions. To verify the performance of the proposed algorithm, zero-waste and non-zero-waste instances are tested. The computational results show that the proposed HAGA is more efficient than the compared algorithms.

The rest of this paper is organized as follows. In Section 2, the mathematical model of 2DRP is built. The improved method, HAGA, for the 2DRP problem is introduced in detail in Section 3. Then, Section 4 provides the computational results and comparisons. Finally, the conclusion of this paper is given in Section 5 with some closing remarks.

## 2. Problem Statement

Assume that there is a set of $n$ rectangle $(R = \{r_1, r_2, \ldots, r_n\})$ items with $w_i \times h_i$ for $i = 1, 2, \ldots, n$. The task is to orthogonally place the items without overlapping in the sheet with $W \times H$ to maximize the filling rate of the sheet, i.e., to maximize the total area of the placed items divided by the area of the sheet. The paper addresses the 2DRP with the following assumptions [30,31]:

(1)　For any two rectangular items put in, they cannot overlap each other.
(2)　When the rectangle is put into the sheet, it cannot exceed the boundary of the sheet.
(3)　The side of the rectangular items must be parallel to the side of the sheet.
(4)　The rectangular can be rotated $90^\circ$.

Based on the problem description and assumptions, a two-dimensional rectangular coordinate system is established with the bottom left corner of the sheet as the origin of the coordinates, and the coordinates of the bottom left corner of the rectangle $r_i$ are $(x_i, y_i)$, as shown in Figure 1. Use variable $g_i$ to indicate whether the rectangle $r_i$ is rotated; $g_i = 0$ means no rotation, and $g_i = 1$ means rotation. If the rectangle $r_i$ is rotated, the width and height of the rectangle are interchanged.

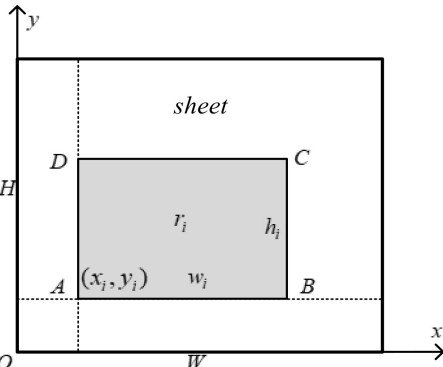

**Figure 1.** Coordinate system.

Therefore, the actual width and height of the rectangle placed change to $w_i(1 - g_i) + h_i g_i$ and $w_i g_i + h_i(1 - g_i)$ respectively, and the coordinates of the top right corner of the rectangle are $(x_i + w_i(1 - g_i) + h_i g_i, y_i + w_i g_i + h_i(1 - g_i))$. It can be seen that the rectangle $r_i$ is represented by $(x_i, y_i, w_i, h_i, g_i)$, which can determine the accurate position of the rectangle in the sheet.

Let $\theta_i$ be a decision variable, taking the value 1 or 0 accordingly as rectangle $r_i$ is or is not placed. The filling rate $F$ of the sheet is defined as the ratio of the total area of the rectangles placed in it to the area of the sheet. The mathematical model of the 2DRP is as follows:

$$\max F = \frac{\sum\limits_{i=1}^{n} w_i h_i \theta_i}{W \times H} \tag{1}$$

s. t.

(1)  $x_i + w_i(1 - g_i) + h_i g_i \leq W, y_i + w_i g_i + h_i(1 - g_i) \leq H$
(2)  $\max\{x_{i1} - x_{j2}, x_{j1} - x_{i2}, y_{i1} - y_{j2}, y_{j1} - y_{i2}\}\theta_i \theta_j \geq 0$
(3)  $(x_{i2} - x_{i1}, y_{i2} - y_{i1}) \in \{(w_i, h_i), (h_i, w_i)\}$
(4)  $\theta_i \in \{0, 1\}, g_i \in \{0, 1\}$
(5)  $0 \leq x_i \leq W, 0 \leq y_i \leq H.$

In constraints (1)–(4), $i, j$ applies to $1, 2, \ldots, n$ and $i \neq j$. Constraint (1) implies that the boundaries of the placed rectangles do not exceed the width and height boundaries of the sheet. Constraint (2) means that the placed rectangles cannot overlap each other. $(x_{i1}, y_{i1})$ and $(x_{i2}, y_{i2})$ represent the coordinates of the bottom left corner and the top right corner of the placed rectangles, respectively. The coordinates of the top right corner can be calculated with the coordinates of the bottom left corner and the width and height of the actual size of the rectangle. Constraint (3) implies that the placed rectangle is parallel to the side of the sheet and can be rotated by $90^\circ$. To sum up, the 2DRP problem can be described as figuring out the best packing solution that satisfies the constraints and maximizes the filling rate of the sheet.

## 3. Improved Adaptive Genetic Algorithm

It is representative for 2DRP to combine heuristic placement strategies and meta-heuristic algorithms such as GA. The GA usually includes SGA and various improved genetic algorithms such as AGA. The SGA is prone to premature problems that cannot escape the local optimal solution. Then, some adaptive genetic algorithms such as AGA [25], AGA [26], and AGA [29] were proposed to solve this problem. However, these also have some problems. Since the AGA [26] is an improvement to the AGA [25], this section mainly focuses on some problems in AGA [26] and AGA [29], presenting further improved AGA for 2DRP by combining heuristic placement strategy.

The packing sequence of rectangles and whether they are rotated are the most important factors that affect the results of the final packing solution. This paper proposes an improved adaptive genetic algorithm as a rectangle sequencing algorithm to determine the packing sequence and rotation state of the rectangles, with the heuristic lowest horizontal line algorithm used as the placement algorithm to determine the position coordinates of the rectangle. First of all, in terms of the genetic algorithm, two-stage encoding is constructed, i.e., the packing sequence is coded by continuous integers from 1 to $n$, and the rotation state is coded with 0 and 1. The initial population is constructed through a combination of sorting rules and random generation. The heuristic lowest horizontal line algorithm is applied to decode, so as to provide the packing solution and calculate the individual fitness. Then, it uses a sort-based method for selection and improved adaptive crossover and mutation operations for evolution. The process of iterative optimization continues until the designed termination rule is reached, and finally, the result decoded by the optimal individual is taken as the final packing solution.

### 3.1. Encoding and Population Initialization

Assuming there are $n$ rectangles $\{r_1, r_2, \ldots, r_n\}$ to be packed, the chromosome adopts a two-stage encoding method consisting of the packing sequence and placement state of the rectangles, and the encoding length is $2n$. The former $n$ uses an integer permutation code, and the packing sequence adopts integer numbers, which is the order of the rectangles to be packed. The latter $n$ uses integer coding, and the rectangle placement status is numbered

with 0 and 1, in which 1 means the placement rectangle rotates, and 0 represents that it does not rotate. So, $s_{n+1}, s_{n+2}, \ldots, s_{n+i}, \ldots, s_{2n}$ are the rotation state of $n$ rectangles to be packed. The overall encoding of individual chromosomes are as follows:

$$E = \underbrace{s_1, s_2, \ldots, s_i, \ldots, s_n}_{n}, \underbrace{s_{n+1}, s_{n+2}, \ldots, s_{n+i}, \ldots, s_{2n}}_{n}$$

where $s_i \in \{1, 2, \ldots, n\}$, $s_{n+i} \in \{0, 1\}$, and $s_i \neq s_j$. For example, there are eight rectangles numbered 1, 2, 3, 4, 5, 6, 7, and 8 to be loaded into the sheet, and each rectangle can be rotated. The packing sequence and rotation status of the two-stage encoding approach for individual chromosome are shown in Figure 2. The order of rectangular packing is 4, 2, 5, 1, 8, 6, 3, and 7, and the rectangles numbered 2, 5, 6, and 7 need to be rotated during the packing process.

| 4 | 2 | 5 | 1 | 8 | 6 | 3 | 7 | 0 | 1 | 1 | 0 | 0 | 1 | 0 | 1 |
|---|---|---|---|---|---|---|---|---|---|---|---|---|---|---|---|

**Figure 2.** Individual chromosome two-stage encoding ($n = 8$).

The above encoding approach accurately describes the factors that affect the packing solution. Considering that the packing sequence sorted by certain attributes tends to be the optimal solution, the following sorting rules [32] are fully utilized to generate the initial population in combination with random generation to speed up the algorithm when initializing the population.

(1)  Sort by area in decreasing order.
(2)  Sort by width in decreasing order.
(3)  Sort by height in decreasing order.
(4)  Sort by perimeter in decreasing order.
(5)  Sort by maximum of width and height in decreasing order.
(6)  Sort by length of diagonal + width + height in decreasing order.

Figure 3 shows the initial population constructed by $PS$ individuals. The individuals from $E_1$ to $E_{ps-6}$ are generated randomly. The packing sequence of the remaining six individuals is generated by the six sorting rules, and the corresponding rotation state of the rectangles is generated randomly. The $PS$ is set to 100 in this paper.

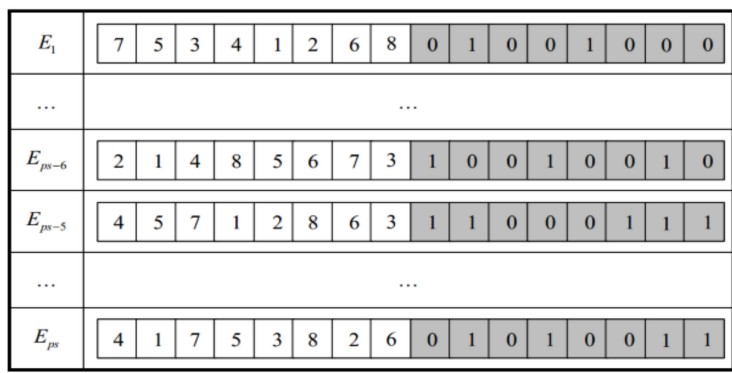

**Figure 3.** Initial population with $PS$ individuals ($n = 8$).

Once the chromosome encoding sequence of the individuals in the population is determined, the heuristic placement algorithm can be used to decode and place, and then the individuals can be evaluated through the fitness function.

### 3.2. Heuristic Placement Algorithm

When using the fitness function to evaluate individuals in the population, it is necessary to arrange the rectangles into the sheet according to a certain heuristic algorithm for

the packing sequence, rotation state, the width and height of rectangles, and the width and height of the sheet to calculate the sheet filling rate and figure out the fitness.

Liu proposed the IBL algorithm [15] to solve the problem that the left side of the BL algorithm [12] is too high and the unfilled cavity area can not be effectively used. Although it can overcome the disadvantage of the BL algorithm, which is prone to the waste area of the sheet, it will lead to the problem of a high right side. In contrast, the lowest horizontal line algorithm proposed in this paper can effectively solve the shortcomings of the BL and IBL algorithms and improve the filling rate [33]. Therefore, the placement algorithm in this paper adopts the heuristic lowest horizontal line algorithm of Ref. [23], and the procedure is as follows in Algorithm 1.

---

**Algorithm 1** Heuristic packing procedure

---

HeuristicPacking(*R, sheet*):
1   Initialize the lowest level line $l$ is 0
2   **while** $R$ is not empty **or** $l$ is beyond the sheet $H$:
3       Get filling area on $l$ and hollow area under $l$, the area size store 2-tuple set (*AW, AH*)
4       Priority to sorting hollow area in increasing order of *AW* for (*AW, AH*)
5       Prepare to place $r_i$ from $R$ in order and rotation state
6       **if** $r_i$ need to rotate:
7           Exchange the width $w_i$ and height $h_i$ of $r_i$
8       **if** $AW \geq w_i$ and $AH \geq h_i$:
9           Place $r_i$, update area size (*AW, AH*)
10          Record $r_i$ the left-bottom corner position coordinates $(x_i, y_i)$,
           placed state, size, rotation state, etc.
11          Remove $r_i$ from $R$
12      **else**:
13          **If** exist the next area:
14              Move the next area to continue placing
15          **else**:
16              Sort these placed $r_i$ in increasing order of $y_i$
17              Update $l$ and the area size (*AL, AH*)
18  **return** the placed information for $R$

---

### 3.3. Fitness Function and Selection Operator

Generally, the fitness function of the genetic algorithm is often determined by the objective function. In this paper, the fitness function refers to the sheet filling rate of the objective function, and its definition can be seen in the 2DRP mathematical model in Section 2.

Selection operation is the process of the genetic algorithm to evaluate the survival of the fittest. Its purpose is to keep the individuals with better fitness in the parent population as much as possible to keep good genes. The traditional roulette [34] selection method gives every individual the opportunity to make a copy, which does not reflect the competitiveness of excellent individuals and cannot realize the principle of survival of the fittest by the genetic algorithm. This paper uses an improved selection method sorted by individual fitness [35] to replace the roulette selection method. The sort-based selection method is described as follows:

Step1. Calculate the fitness of each individual in the population.

Step2. Sort the individuals in the population in descending order of fitness.

Step3. Divide the sorted individuals into three parts, and the first one was duplicated into two copies.

Step4. Make a copy of the individuals in the middle.

Step5. The remaining individuals ranked behind are not copied.

### 3.4. Adaptive Crossover Operator and Mutation Operator

This subsection gives an improved genetic algorithm, in which the crossover probability and mutation probability can be adjusted adaptively. The crossover operation is to randomly exchange some genes between two individuals in the population based on the crossover probability so as to combine excellent genes to produce new and better individuals, which is the main part of the genetic algorithm. The mutation operation is to replace genes of an individual with other alleles under a certain mutation probability, thereby forming a new individual, ensuring the diversity of the population and preventing the phenomenon of premature.

Crossover probability and mutation probability are the most important parameters for crossover and mutation operations. Their selection is the key to influencing the behavior and performance of the algorithm and directly affecting the convergence of the algorithm. Regarding the crossover probability, if the crossover probability is too small, the search process will be slow and will easily cause stagnation. However, if the crossover probability is too large, the chromosomal structure of an individual with high fitness will be quickly destroyed and replaced. For the mutation probability, if it is too small, a new individual chromosome structure is not easy to be generated, and the search space will become narrower. Otherwise, if the mutation probability is too large, the genetic algorithm becomes a purely random search algorithm, which makes it easy to fall into a local optimal solution.

The AGA [26] proposed by Ren is to adaptively change the crossover probability and mutation probability from the local point according to individual fitness without considering whether the dynamic adjustment of crossover and mutation has a positive impact on population evolution from the whole point of population evolution. The AGA [29] proposed by Jiang dynamically adjusts the crossover probability and mutation probability according to the fitness changes of the optimal individual from the whole point of the evolution of the population offspring. However, the crossover probability and mutation probability of all individuals of a certain generation of population are still the same, and the performance of individuals with different fitness of the generation is consistent, so there is no difference. Moreover, the AGA [29] is greatly affected by the contemporary optimal fitness and the range of fitness function, which will limit the wide application of the algorithm.

Inspired by Ren and Jiang, this paper proposes an improved method called hybrid adaptive genetic algorithm (HAGA) to solve the 2DRP problem. The adaptive description of the crossover probability is as follows:

$$P_c(t) = P_{c2} + (P_{c1} - P_{c2})\frac{1 - e^{-ct}}{1 + e^{-ct}}, t \geq 0 \tag{2}$$

$$P_c(f') = \begin{cases} P_{c1} - \frac{(P_{c1} - P_{c2})(f' - f_{avg})}{f_{\max} - f_{avg}} &, f' \geq f_{avg} \\ P_{c1} &, f' < f_{avg} \end{cases} \tag{3}$$

$$P_c = \lambda P_c(t) + (1 - \lambda)P_c(f') \tag{4}$$

where $P_{c1}$ and $P_{c2}$ are the upper bound and lower bound of $P_c$. In general, the recommended parameters for these two crossover probabilities are 0.9 and 0.6 respectively [36]; i.e., $P_{c1} = 0.9$ and $P_{c2} = 0.6$. The parameter $t$ is the cumulative number of generations in which the optimal fitness value of the population has not changed. When the optimal fitness value of a certain generation changes, it will be set to 0 for re-accumulation. For example, in the process of iterative optimization, if the best fitness value lasts for ten times without changing, the $t$ is set to 10. The parameter $c$ is a constant to determine the slope of the $(1 - e^{-ct})/(1 + e^{-ct})$ function, and $c = 0.2$ in the experiment. $f'$ is the fitness value of the larger individuals that are ready to implement crossover; $f_{\max}$ is the maximum fitness value of the current population; $f_{avg}$ is the average fitness value of the current population. The $\lambda$ is an adjustable weight parameter with the range of $0 < \lambda < 1$, and $\lambda = 0.6$ in the experiment.

$(1 - e^{-ct})/(1 + e^{-ct})$ in Formula (2) has a value range of 0–1 when $t$ is greater than or equal to 0, which ensures that $P_c(t)$ changes between $P_{c2}$ and $P_{c1}$. It also helps to solve the problem for AGA [29] that is greatly affected by the contemporary optimal fitness, representing the fitness function range with appropriately adjusting $P_{c1}$ and $P_{c2}$. It is beneficial to combine with AGA [26]. The $t$ is recorded after each iteration to determine the change of $P_c(t)$. If $t$ becomes larger, it shows that the optimal fitness of the population evolution has not changed, and the crossover probability needs to be increased to promote the change of the optimal fitness. If $t = 0$, it reveals that the optimal fitness of the population evolution has increased, and the crossover probability $P_c(t)$ will drop to $P_{c2}$ for increasing more possible solutions. Then, combining Formula (3) to increase the difference of crossover operation for different individuals. Combining Formula (2) and Formula (3), it works together to find a better solution from the global perspective of population evolution and the local perspective of individual fitness. In addition, because it can increase the crossover probability from the perspective of the population evolution process, it is also helpful to overcome the problem that evolution is likely to stagnate when $f_{avg}$ is close to $f_{max}$. The effect of the adaptive mutation probability change in the latter part is similar to this.

When carrying out crossover, the packing sequence gene string adopts the partial matching crossover (PMX), and the rotating state gene string adopts the two-point crossover method. An example of the crossover process is shown in Figure 4. The procedure of the hybrid adaptive crossover algorithm is described in Algorithm 2.

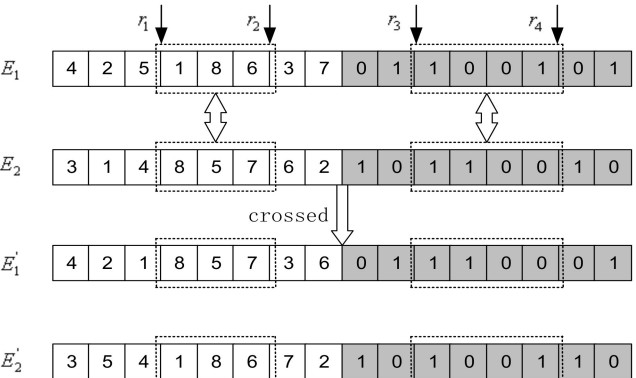

**Figure 4.** Partial matching crossover (PMX) and two-point crossover ($n = 8, r_1 = 4, r_2 = 6, r_3 = 11, r_4 = 14$).

---

**Algorithm 2** Hybrid adaptive crossover algorithm

---

HACA($t, P, PS$):
1   Initialize the parameter $P_{c1}, P_{c2}, c, \lambda$
2   Copy population $P'$ from the population $P$ with $PS$ individuals
3   Compute $P_c(t)$ with formula (2)
4   Compute $f_{max}, f_{avg}$ for $P$
5   **for** $i$ **in** range(0, $PS$, 2):
6      Randomly select two different parents $E_1, E_2$ from $P$
7      Compute greater fitness $f'$ from $E_1, E_2$
8      Compute $P_c(f')$ with formula (3) and $P_c$ with formula (4)
9      Generate a random number $r$ between [0, 1] with rand() function
10     **if** $r < P_c$:
11        Generate different random numbers $r_1, r_2$ within $[1, n]$ and $r_3, r_4$ within $[n + 1, 2n]$
12        Get two crossed children using the same method of Figure 4
13        Put two crossed children with Heuristic Packing Procedure
14        Update $P'[i], P'[i + 1]$ using two crossed children
15  Add $P'$ to $P$, Sort them in decreasing order of fitness
16  Delete $PS$ individuals with small fitness and get crossed population $P$
17  **return** crossed population $P$

---

The adaptive description of mutation probability is as follows:

$$P_m(t) = P_{m2} + (P_{m1} - P_{m2})\frac{1 - e^{-ct}}{1 + e^{-ct}}, t \geq 0 \tag{5}$$

$$P_m(f) = \begin{cases} P_{m1} - \frac{(P_{m1} - P_{m2})(f - f_{avg})}{f_{\max} - f_{avg}} & , f \geq f_{avg} \\ P_{m1} & , f < f_{avg} \end{cases} \tag{6}$$

$$P_m = \lambda P_m(t) + (1 - \lambda)P_m(f) \tag{7}$$

where $P_{m1}$ and $P_{m2}$ are the upper bound and lower bound of $P_m$. In the experiment, these two mutation probability parameters recommended are 0.1 and 0.5 respectively [36]; i.e., $P_{m1} = 0.5$ and $P_{m2} = 0.1$. $f$ is the fitness value of the individual to be mutated; $f_{\max}$ is the maximum fitness value of the current population; and $f_{avg}$ is the average fitness value of the current population. For other parameters, refer to the section on adaptive crossover probability. Similarly, we can be aware that Formula (5) is the adaptive mutation probability from the whole point of population offspring evolution, and Formula (6) is the adaptive mutation probability for individual fitness from a local perspective. The hybrid adaptive mutation probability is used to work together to dynamically adjust the mutation probability.

When mutating, the packing sequence gene string adopts the exchange mutation method, and the rotating state gene string takes the two-point basic position mutation method. An example of the mutation process is shown in Figure 5.

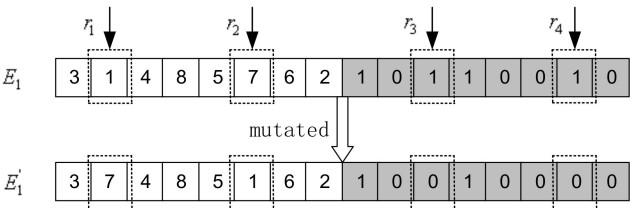

**Figure 5.** Exchange mutation and two-point basic position mutation ($n = 8, r_1 = 2, r_2 = 6$, $r_3 = 11, r_4 = 15$).

The procedure of the hybrid adaptive mutation algorithm is described in Algorithm 3. In Algorithms 2 and 3, $P$ represents the population in the genetic algorithm; $PS$ means the size of the population, i.e., the number of individuals in the population; The parameter $t$ is obtained by comparing the fitness of the best individual in the previous and next generations in the population evolution process. If the optimal individual fitness of the previous generation and the next generation are equal, $t = t + 1$; otherwise, $t$ is reset to 0.

In summary, Algorithm 2 plays a role of constructing the hybrid adaptive crossover algorithm (HACA) and Algorithm 3 structures the hybrid adaptive mutation algorithm (HAMA), which are two main parts of HAGA. From the local aspect of individual fitness and the whole evolution of later generations' population, the crossover probability and mutation probability can be adjusted adaptively to quickly find a further improved solution.

### 3.5. Termination Rules

The improved adaptive genetic algorithm we proposed shows a convergence trend in the process of searching and optimizing the solution space. For the iterative optimization calculation of 2DRP problem, there are the following termination rules:

(1) After the iterative calculation reaches a certain level, the optimal fitness value has no obvious change in the continuous $T$ iterations, and the calculation can be stopped in advance. For example, the $T$ is set to 150 in this paper, which means the calculation is terminated if the optimal fitness value does not change through 150 continuous iterations;

(2) Set the desired packing rate in advance. In the process of calculating, the pre-set filling rate is reached, and the calculation can be terminated early. For the expected filling

rate, if the theoretical filling rate is known before the experiment, e.g., the theoretical filling rate of the *C* benchmark instances proposed by Hopper is 100% [18], it can be set to the theoretical value; otherwise, the user's expected value can be set according to actual needs;

(3) Set the maximum number of iterations in advance. If the algorithm does not terminate prematurely, when the genetic algebra reaches the maximum number of iterations, the calculation is terminated, and the packing solution and filling rate are obtained.

The combination of the above three termination rules can not only fully optimize within the maximum number of iterations but also terminate the iterative optimization of the algorithm early when the expected value or fitness value is reached and there is no obvious change, which is beneficial to reduce the calculation time of the algorithm. When the iterative optimization of the algorithm is terminated, the packing solution corresponding to the chromosome with the largest fitness value is the optimal packing solution, and the corresponding filling rate is calculated.

---

**Algorithm 3** Hybrid adaptive mutation algorithm

---

HAMA($t, P, PS$):
1   Initialize the parameter $P_{m1}, P_{m2}, c, \lambda$
2   Copy population $P'$ from the population $P$ with $PS$ individuals
3   Compute $P_m(t)$ with formula (5)
4   Compute $f_{\max}, f_{avg}$ for $P$
5   **for** $i$ **in** range($PS$):
6      Get parent $E_1$ from $P$ and compute its fitness $f$
7      Compute $P_m(f)$ with formula (6) and $P_m$ with formula (7)
8      Generate a random number $r$ between [0, 1] with rand() function
9      **if** $r < P_m$:
10         Generate different random numbers $r_1, r_2$ within $[1, n]$ and $r_3, r_4$ within $[n+1, 2n]$
11         Get mutated child using the same method of Figure 5
12         Put mutated child with Heuristic Packing Procedure
13         Update $P'[i]$ using mutated child
14   Add $P'$ to $P$, Sort them in decreasing order of fitness
15   Delete $PS$ individuals with small fitness and get mutated population $P$
16   **return** mutated population $P$

---

### 3.6. Related Discussion

The proposed algorithm firstly overcomes the problem that is greatly affected by the contemporary optimal fitness and fitness function range for AGA [29] to expand its application; especially, the fitness function range is not between 0 and 1. Moreover, this algorithm not only obtains the differential performance of a certain generation of individuals by combining AGA [26] but also handles the impact of dynamic changes on population evolution to find adaptively an improved solution. However, the presented algorithm is not perfect in this paper; it has the following drawbacks:

(1) There are some parameters that need to be controlled, including $P_{c1}, P_{c2}, P_{m1}, P_{m2}, \lambda$, and $c$. The $P_{c1}, P_{c2}, P_{m1}$, and $P_{m2}$ may be obtained through existing experience. It is necessary to do sets of comparative experiments to establish consistent parameters $\lambda, c$ to obtain good benefits.

(2) From Formulas (2) and (5), if $t = 0$, it can be concluded that $P_c(t)$ suddenly drops to the minimum of crossover probability and mutation probability $P_{c2}$ and $P_{m2}$. In the early stage of evolution, the optimal individual is easy to change frequently; that is, there are more times where $t = 0$, which leads to lower crossover probability and mutation probability that are not beneficial to rapid population evolution.

We can set up a series $\lambda$ and $c$ to determine the appropriate parameters through many experiments. In addition, it is possible for $t = 0$ to make continuous descent changes for different evolutionary periods.

## 4. Computational Experiments

In this section, to evaluate the effectiveness of the proposed improved method, HAGA, four computational experiments are designed and implemented. The rectangle packing instances, which include zero-waste and non-zero-waste datasets, are used for testing. The first three benchmark instances as zero-waste are the *J* instances ($J1 - J2$) proposed by Jakobs [37], the *C* instances ($C11 - C73$) proposed by Hopper et al. [18], and the *N* instances ($N1 - N13$) proposed by Burke et al. [19]. The scale and optimization difficulty of these three benchmark instances increase sequentially. The optimal solution for each instance is known beforehand, in which the total area of the given items is equivalent to the area of the sheet, resulting in a 100% filling rate. So, they all theoretically have a 100% filling rate but are not balanced dimensionally. However, considering that non-zero-waste problems often occur in real application, the non-zero-waste datasets, including 4 instances (cgcut1-cgcut3) given in Ref. [38], 13 instances (gcut1-gcut13) in Ref. [39], 12 instances (ngcut1-ngcut12) in Ref. [40], and 10 instances (beng1-beng10) in Ref. [41], are used for further testing.

The SGA, AGA [26], AGA [29], and HAGA involved in the experiment are implemented in Python language programming and calculated on an Intel Core i5 CPU at 2.5 GHz with 8 GB memory. We tested the effects of SGA and AGA [26] for each individual fitness proposed by Ren, AGA [29] for the evolutionary process proposed by Jiang, and the HAGA proposed in this paper. Then, a comparison of HAGA with some recent algorithms is provided. In order to balance the calculation accuracy and efficiency of the test algorithm, the number of iterations $NIND$ is set to 2000, and the calculation result is the average of twenty operations.

### 4.1. Computational Experiment for Zero-Waste Instances

#### 4.1.1. Computational Experiment for *J* Instances

We know that Ref. [42] divides the *J* benchmark instances into two groups of problem instances, which are called *J*1 and *J*2. The size of the large rectangular sheet is $40 \times 15$. The number of items in the first group is 25, and the number of items in the second group is 50. For more detailed information on the instances, refer to the original paper published by Jakobs [37].

In this subsection, we compare the performance of the proposed HAGA with SGA, AGA [26], and AGA [29] on the *J* instances. The AGA [26] is an adaptive genetic algorithm that dynamically adjusts the crossover probability and mutation probability for each individual. The AGA [29] is an adaptive genetic algorithm that dynamically adjusts the crossover probability and mutation probability for the evolution process. In the comparative experiment, the crossover probability and mutation probability in SGA are 0.9 and 0.1, respectively. The parameters $P_{c1}$, $P_{c2}$, $P_{m1}$ and $P_{m2}$ of the AGA [26] remain consistent with that in this paper, and the parameter $c$ in AGA [29] also remains consistent with this paper. The computational results are shown in Table 1, where F represents the filling rate of the sheet. Figures 6 and 7 show the rectangular packing layouts corresponding to the optimal solution obtained by the HAGA.

**Table 1.** Comparison of the filling rate (%) and calculation time (s) on the *J* instances.

| Instances | $n$ | Sheet | SGA | | AGA [26] | | AGA [29] | | HAGA | |
|---|---|---|---|---|---|---|---|---|---|---|
| | | | F/% | Time/s | F/% | Time/s | F/% | Time/s | F/% | Time/s |
| J1 | 25 | $40 \times 15$ | 100 | 10.71 | 100 | 8.37 | 100 | 7.69 | 100 | 5.59 |
| J2 | 50 | $40 \times 15$ | 99.65 | 27.18 | 99.875 | 24.53 | 99.90 | 25.11 | 99.917 | 23.3 |
| | Average | | 99.825 | 18.95 | 99.94 | 16.45 | 99.95 | 16.4 | 99.96 | 14.45 |

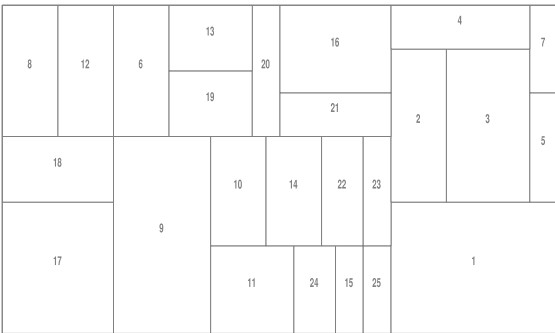

**Figure 6.** Full packing layouts on *J*1 instance.

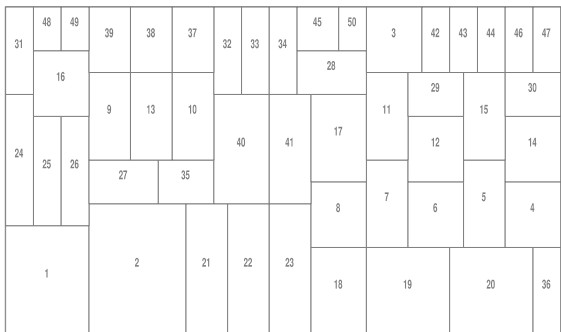

**Figure 7.** Full packing layouts on *J*2 instance.

From the results shown in Table 1, it can be seen that the average calculation time of the three adaptive genetic algorithms of AGA [26], AGA [29], and HAGA are 16.45 s, 16.4 s, and 14.45 s respectively, which is less than that of the SGA. It shows that the adaptive genetic algorithm can accelerate the convergence speed of the algorithm by dynamically adjusting the crossover probability and mutation probability. The average filling rate of the adaptive genetic algorithm is higher than that of SGA, which shows that the adaptive genetic algorithm can address the problem that SGA may fall into a local optimal solution. Next, by comparing three adaptive genetic algorithms, we can realize that they have almost the same average filling rate on the *J* instances, but the average time consumed for HAGA is 2 s, which is less than AGA [26] and 1.95 s less than AGA [29]. The results show that the HAGA can overcome the shortcomings of the two types of adaptive genetic algorithms including AGA [26] and AGA [29] through the joint effect of the local aspects of individual and the overall aspects of evolution, and the convergence speed of the algorithm is accelerated to show the effectiveness of the proposed algorithm.

### 4.1.2. Computational Experiment for *C* Instances

The number of items per instance for *C* benchmark instances ranges from 16 to 197. According to the area of the sheet, these instances can be classified into seven groups with three instances each. For detailed information on the *C* instances, refer to Hopper et al. [18]. The experimental comparison results are shown in Table 2.

**Table 2.** Comparison of the filling rate (%) and calculation time (s) on the *C* instances.

| Instances | $n$ | Sheet | SGA | | AGA [26] | | AGA [29] | | HAGA | |
|---|---|---|---|---|---|---|---|---|---|---|
| | | | F/% | Time/s | F/% | Time/s | F/% | Time/s | F/% | Time/s |
| C11 | 16 | 20 × 20 | 100 | 2.41 | 100 | 2.14 | 100 | 2.16 | 100 | 1.64 |
| C12 | 17 | 20 × 20 | 100 | 2.49 | 100 | 2.33 | 100 | 2.43 | 100 | 1.7 |
| C13 | 16 | 20 × 20 | 100 | 2.28 | 99.65 | 2.14 | 100 | 2.1 | 100 | 1.62 |
| C21 | 25 | 40 × 15 | 100 | 10.83 | 100 | 8.55 | 100 | 7.79 | 100 | 5.54 |
| C22 | 25 | 40 × 15 | 100 | 9.03 | 100 | 7.87 | 100 | 7.41 | 100 | 5.22 |
| C23 | 25 | 40 × 15 | 100 | 9.19 | 100 | 7.58 | 100 | 7.23 | 100 | 5.37 |
| C31 | 28 | 60 × 30 | 100 | 13.41 | 100 | 11.82 | 100 | 10.72 | 100 | 8.92 |
| C32 | 29 | 60 × 30 | 100 | 13.73 | 100 | 11.54 | 100 | 10.94 | 100 | 9.12 |
| C33 | 28 | 60 × 30 | 100 | 12.81 | 100 | 10.81 | 100 | 10.66 | 100 | 8.86 |
| C41 | 49 | 60 × 60 | 100 | 22.52 | 100 | 21.94 | 100 | 21.07 | 100 | 19.8 |
| C42 | 49 | 60 × 60 | 100 | 23.28 | 100 | 21.55 | 100 | 20.9 | 100 | 19.17 |
| C43 | 49 | 60 × 60 | 100 | 22.37 | 100 | 21.87 | 100 | 21.32 | 100 | 20.14 |
| C51 | 73 | 60 × 90 | 99 | 338.13 | 100 | 46.23 | 100 | 70.11 | 100 | 41.82 |
| C52 | 73 | 60 × 60 | 100 | 57.46 | 100 | 42.53 | 100 | 40.27 | 100 | 38.31 |
| C53 | 73 | 60 × 60 | 99.62 | 327.17 | 100 | 44.07 | 100 | 68.44 | 100 | 42.19 |
| C61 | 97 | 80 × 120 | 98.69 | 483.98 | 100 | 127.85 | 100 | 106.97 | 100 | 100.85 |
| C62 | 97 | 80 × 120 | 100 | 243.48 | 100 | 53.43 | 100 | 132.54 | 100 | 54.78 |
| C63 | 97 | 80 × 120 | 100 | 386.6 | 100 | 111.63 | 100 | 187.37 | 100 | 105.42 |
| C71 | 196 | 160 × 240 | 98.43 | 1686.13 | 98.94 | 1643.48 | 99.08 | 1603.96 | 99.52 | 1551.02 |
| C72 | 197 | 160 × 240 | 98.67 | 1825.58 | 98.97 | 1779.65 | 99.23 | 1748.54 | 100 | 1501.37 |
| C73 | 196 | 160 × 240 | 98.34 | 1672.37 | 98.82 | 1619.88 | 98.94 | 1572.38 | 100 | 1335.34 |
| | Average | | 99.65 | 341.2 | 99.83 | 266.61 | 99.87 | 269.3 | 99.98 | 232.29 |

It can be inferred from the results in Table 2 that as the scale *n* of the problem instances becomes larger and larger, the difficulty of the algorithm convergence increases, and some instances cannot achieve a 100% filling rate. The HAGA can obtain a 100% filling rate for all instances except for C71, but AGA [26] fails to obtain a 100% filling rate for C13, C71, C72, and C73. The scale of the C13 instance is not large, but the 100% filling rate is not obtained because the algorithm fell into a possible local optimal solution during a certain test. In addition, the AGA [29] fails to obtain a 100% filling rate for C71, C72, and C73. It is obvious that the HAGA is better than AGA [26] and AGA [29].

Then, by comparing SGA and the adaptive genetic algorithm, it can be realized that AGA [26], AGA [29], and HAGA save 74.59, 71.9, and 108.91 s respectively in comparison to SGA. The average filling rate of the adaptive genetic algorithm is generally higher than that of SGA, which shows that the adaptive genetic algorithm can overcome the "premature" phenomenon to which SGA is prone. Through the analysis of the results of the three adaptive genetic algorithms, it can be found that the HAGA uses 34.32 s and 37.01 s less than AGA [26] and AGA [29] in the average calculation time, and it increases the average filling rate by 0.15% and 0.11%, respectively. This shows that the improved method, HAGA, can speed up the convergence of the algorithm and relatively increase the filling rate of the sheet.

#### 4.1.3. Computational Experiment for *N* Instances

The *N* benchmark instances were generated by Burke et al., and optimal solutions were also known beforehand. It includes 13 groups of test instances, and the number of items to be packed increases from 10 in the first group to 3152 in the thirteenth group. For detailed information on *N* instances, refer to Burke et al. [19]. The comparison results of our experiments are shown in Table 3. It should be noted that the '-' in N13 means that no experimental calculation has been performed.

We can recognize from the results shown in Table 3 that SGA can only obtain a 100% filling rate for N1–N3 instances with a small scale. Both AGA [26] and AGA [29] fail to obtain a 100% filling rate in five instances, and HAGA fails to obtain a 100% filling rate in three instances N10~N12. Comparing the N5 and C41–C43 instances of almost the

same scale, you will find that the N5 instance consumes a lot more time than the C41–C43 instances. For example, the calculation time of AGA [26] is 22.37 s on the C43 instance, but it takes 33.98 s on the N5 instance. These indicate that solving *N* instances is more difficult than solving the *C* instances, but the HAGA is still better than the two types of adaptive methods AGA [26] and AGA [29] to solve the 2DRP problem.

Then, by comparing SGA and the adaptive genetic algorithm, we figure out that AGA [26], AGA [29], and HAGA reduce the time by 90.46, 114.08, and 154.49 s, respectively compared with SGA, and the average filling rate of the adaptive genetic algorithm is generally higher than that of SGA. We compare the computational results of the three adaptive genetic algorithms again, and it can be seen that the HAGA saves 64.03 s and 40.41 s in the average calculation time than AGA [26] and AGA [29], and it increases the average filling rate by 0.1% and 0.05%, respectively. Therefore, it further shows that the HAGA can better improve the convergence speed and filling rate for 2DRP, thereby improving the packing efficiency and quality.

### 4.2. Computational Experiment for Non-Zero-Waste Instances

The 38 non-zero-waste instances including cgcut1-cgcut3, gcut1-gcut13, ngcut1-ngcut12, and beng1-beng10 are evaluated to further verify the effectiveness of the HAGA. These datasets have been used by other authors to verify the performance of the algorithm. Although some of the area of the stock sheet is wasted in these instances, the optimal solution of problem can be known from Ref. [43]. It should be noted that the filling rate needs to be obtained through the $(1 - trim\ loss) \times 100$ formula conversion for the trim loss indicator in Ref. [43]. The data can be downloaded from http://people.brunel.ac.uk/~mastjjb/jeb/info.html.

Table 4 reports the comparison results of AGA [26], AGA [29], and HAGA for non-zero-waste instances. From Table 4, we can observe that the number of instances in which HAGA, AGA [26], and AGA [29] obtained the optimal filling rate is 22, 20, and 21, respectively. The HAGA obtains the optimal filling rate except for cgcut3 on the cgcut1-cgcut3 instances, ngcut6, ngcut9, and ngcut11 on the ngcut1-gcut12 instances, and beng10 on the beng1-beng10 instances. The worst performance of HAGA is that it only obtains the optimal filling rate on gcut1 and gcut3 for the gcut1-gcut13 instances. The filling rate of HAGA is not lower than that of AGA [26] except for five instances (gcut5, gcut9, gcut11, gcut12, ngcut11). Similarly, the filling rate of HAGA is not lower than that of AGA [29] except for six instances (gcut2, gcut7, gcut9, gcut10, ngcut12, beng10). By comparing the computational results of the three adaptive genetic algorithms, it can be observed that the HAGA saves 19.5 s and 15.8 s in average calculation time compared with AGA [26] and AGA [29], and it improves the average filling rate by 0.2% and 0.06%, respectively. The computational results show that the HAGA is still effective for non-zero-waste instances.

**Table 3.** Comparison of the filling rate (%) and calculation time (s) on the *N* instances.

| Instances | *n* | Sheet | SGA | | AGA [26] | | AGA [29] | | HAGA | |
|---|---|---|---|---|---|---|---|---|---|---|
| | | | F/% | Time/s | F/% | Time/s | F/% | Time/s | F/% | Time/s |
| N1 | 10 | 40 × 40 | 100 | 1.57 | 100 | 1.23 | 100 | 1.3 | 100 | 0.94 |
| N2 | 20 | 30 × 50 | 100 | 6.89 | 100 | 4.65 | 100 | 4.23 | 100 | 3.37 |
| N3 | 30 | 30 × 50 | 100 | 14.31 | 100 | 11.72 | 100 | 10.86 | 100 | 9.46 |
| N4 | 40 | 80 × 80 | 99.75 | 56.63 | 100 | 20.59 | 100 | 19.55 | 100 | 16.9 |
| N5 | 50 | 100 × 100 | 99.4 | 77.01 | 100 | 33.98 | 99.93 | 60.8 | 100 | 27.61 |
| N6 | 60 | 50 × 40 | 99.64 | 152.57 | 100 | 135.93 | 100 | 123.56 | 100 | 41.2 |
| N7 | 70 | 80 × 100 | 99.45 | 207.05 | 99.86 | 184.59 | 99.95 | 158.22 | 100 | 114.96 |
| N8 | 80 | 100 × 80 | 99.16 | 227.67 | 99.89 | 190.09 | 100 | 92.47 | 100 | 47.28 |
| N9 | 100 | 50 × 150 | 98.76 | 323.84 | 100 | 224.58 | 100 | 183.23 | 100 | 131.4 |
| N10 | 200 | 70 × 150 | 98.63 | 1582.76 | 99.09 | 1454.21 | 99.28 | 1433.97 | 99.23 | 1378.05 |
| N11 | 300 | 50 × 150 | 98.39 | 2998.58 | 98.76 | 2728.4 | 98.87 | 2713.61 | 99.08 | 2724.25 |
| N12 | 500 | 100 × 300 | 97.67 | 7170.08 | 98.02 | 6743.56 | 98.26 | 6648.27 | 98.54 | 6469.66 |
| N13 | 3152 | 640 × 960 | - | - | - | - | - | - | - | - |
| | Average | | 99.24 | 1068.25 | 99.64 | 977.79 | 99.69 | 954.17 | 99.74 | 913.76 |

**Table 4.** Comparison of the filling rate (%) and calculation time (s) on the non-zero-waste instances.

| Instances | *n* | W | H | Opt. | AGA [26] | | AGA [29] | | HAGA | |
|---|---|---|---|---|---|---|---|---|---|---|
| | | | | | F/% | Time/s | F/% | Time/s | F/% | Time/s |
| cgcut1 | 16 | 10 | 23 | 97.8261 | 97.8261 | 2.4 | 97.8261 | 2.23 | 97.8261 | 1.31 |
| cgcut2 | 23 | 70 | 63 | 98.5034 | 98.5034 | 6.33 | 98.5034 | 7.19 | 98.5034 | 4.17 |
| cgcut3 | 62 | 70 | 636 | 99.9551 | 95.5931 | 291.99 | 95.5034 | 297.26 | 96.3971 | 293.18 |
| gcut1 | 10 | 250 | 1016 | 64.3945 | 64.3945 | 1.09 | 64.3945 | 0.94 | 64.3945 | 0.27 |
| gcut2 | 20 | 250 | 1133 | 96.9331 | 90.1407 | 78.9 | 91.7602 | 74.17 | 91.5223 | 72.21 |
| gcut3 | 30 | 250 | 1803 | 90.4384 | 90.4384 | 117.58 | 90.4384 | 119.25 | 90.4384 | 111.95 |
| gcut4 | 50 | 250 | 2934 | 99.7148 | 93.3675 | 217.78 | 94.1787 | 220.15 | 94.8336 | 212.34 |
| gcut5 | 10 | 500 | 1172 | 93.0546 | 85.7871 | 33.64 | 87.1279 | 36.93 | 84.3447 | 33.86 |
| gcut6 | 20 | 500 | 2514 | 98.0157 | 93.2039 | 70.29 | 93.5469 | 71.78 | 94.3003 | 69.82 |
| gcut7 | 30 | 500 | 4641 | 86.3948 | 84.1737 | 117.04 | 84.7101 | 119.29 | 84.5164 | 119.01 |
| gcut8 | 50 | 500 | 5703 | 98.3855 | 94.121 | 225.67 | 94.4197 | 219.42 | 94.7485 | 215.17 |
| gcut9 | 10 | 1000 | 2022 | 99.9916 | 95.1696 | 33.89 | 94.2799 | 41.33 | 92.6305 | 34.77 |
| gcut10 | 20 | 1000 | 5356 | 99.9884 | 91.2531 | 73.14 | 91.9732 | 77.88 | 91.6382 | 71.28 |
| gcut11 | 30 | 1000 | 6537 | 99.9927 | 94.7406 | 112.19 | 93.6346 | 116.25 | 93.9769 | 113.58 |
| gcut12 | 50 | 1000 | 12522 | 99.9999 | 92.012 | 216.88 | 92.3138 | 210.78 | 91.4614 | 227.23 |
| gcut13 | 32 | 1000 | 4772 | 99.9966 | 94.5067 | 143.68 | 94.2138 | 150.77 | 96.1827 | 127.13 |
| ngcut1 | 10 | 10 | 23 | 82.6087 | 82.6087 | 1.18 | 82.6087 | 0.87 | 82.6087 | 0.26 |
| ngcut2 | 17 | 10 | 30 | 92.3333 | 92.3333 | 3.89 | 92.3333 | 3.27 | 92.3333 | 2.34 |
| ngcut3 | 21 | 10 | 28 | 98.9286 | 98.9286 | 4.25 | 98.9286 | 4.66 | 98.9286 | 2.99 |
| ngcut4 | 7 | 10 | 20 | 81.0000 | 81.0000 | 0.69 | 81.0000 | 0.5 | 81.0000 | 0.16 |
| ngcut5 | 14 | 10 | 36 | 98.0556 | 98.0556 | 48.57 | 98.0556 | 50.41 | 98.0556 | 45.2 |
| ngcut6 | 15 | 10 | 29 | 100.000 | 93.7034 | 61.22 | 94.3448 | 53.94 | 94.8276 | 53.59 |
| ngcut7 | 8 | 20 | 20 | 43.7500 | 43.7500 | 0.39 | 43.7500 | 0.34 | 43.7500 | 0.2 |
| ngcut8 | 13 | 20 | 32 | 98.9063 | 98.9063 | 45.8 | 98.9063 | 43.32 | 98.9063 | 5.97 |
| ngcut9 | 18 | 20 | 49 | 99.3878 | 95.9147 | 61.62 | 96.1224 | 63.4 | 97.3061 | 31.49 |
| ngcut10 | 13 | 30 | 80 | 71.6667 | 71.6667 | 0.64 | 71.6667 | 0.6 | 71.6667 | 0.29 |
| ngcut11 | 15 | 30 | 50 | 98.8667 | 92.3333 | 51.73 | 91.0667 | 46.99 | 91.3067 | 37.85 |
| ngcut12 | 22 | 30 | 87 | 87.9693 | 85.6705 | 81.68 | 87.4866 | 76.49 | 87.9693 | 40.19 |
| beng1 | 20 | 25 | 30 | 98.8000 | 98.8000 | 4.83 | 98.8000 | 4.67 | 98.8000 | 3.19 |
| beng2 | 40 | 25 | 57 | 99.6491 | 99.6491 | 21.58 | 99.6491 | 22.49 | 99.6491 | 16.42 |
| beng3 | 60 | 25 | 84 | 99.5238 | 99.5238 | 84.97 | 99.5238 | 73.34 | 99.5238 | 49.58 |
| beng4 | 80 | 25 | 107 | 99.9252 | 99.9252 | 30.63 | 99.9252 | 34.04 | 99.9252 | 20.39 |
| beng5 | 100 | 25 | 134 | 99.4030 | 99.4030 | 196.13 | 99.4030 | 228.56 | 99.4030 | 116.13 |
| beng6 | 40 | 40 | 36 | 98.6111 | 98.6111 | 10.84 | 98.6111 | 14.92 | 98.6111 | 12.6 |
| beng7 | 80 | 40 | 67 | 99.7388 | 99.7388 | 53.03 | 99.7388 | 42.23 | 99.7388 | 24.97 |

**Table 4.** *Cont.*

| Instances | $n$ | W | H | Opt. | AGA [26] | | AGA [29] | | HAGA | |
|---|---|---|---|---|---|---|---|---|---|---|
| | | | | | F/% | Time/s | F/% | Time/s | F/% | Time/s |
| beng8 | 120 | 40 | 101 | 99.6782 | 98.9697 | 184.26 | 98.8132 | 193.07 | 99.6782 | 123.85 |
| beng9 | 160 | 40 | 126 | 99.3651 | 98.8396 | 1063.11 | 99.3651 | 880.52 | 99.3651 | 735.11 |
| beng10 | 200 | 40 | 156 | 99.6314 | 98.6238 | 1197.35 | 98.7903 | 1206.47 | 98.7342 | 1178.83 |
| | | Average | | | 91.6365 | 130.29 | 91.7819 | 126.6 | 91.8369 | 110.76 |

*4.3. Compare the Results with Other Algorithms*

We compare the proposed algorithm with the algorithms in Ref. [43–45]. A hybrid simulated annealing algorithm (HSA) proposed by Leung et al. [43] uses a greedy strategy and simulated annealing to obtain an improved solution. A least-waste-first heuristic algorithm (LWF) proposed by Wei et al. [44] presents a least wasted first heuristic algorithm to find the desirable solution. An iterative bidirectional heuristic placement (IBHP) proposed by Shiangjen et al. [45] combines the constructive heuristic algorithms of bidirectional heuristic placement (BHP) and iterated local search (ILS) with a shift strategy to improve the solution. Considering that Ref. [44] lacks some test instances, the results of LWF and HSA are taken directly from Ref. [43]. The HSA and LWF in C++ programming language were run on a Dell GX270 with 2.60 GHz CPU and 512 MB memory. The IBHP in C programming language was performed on an Intel Core i5 CPU at 2.90 GHz with 4 GB of RAM running on the Ubuntu 12.04 operating system. The comparisons of the average filling rate and the average calculation time between LWF, HSA, IBHP, and HAGA are shown in Table 5. The '-' in Table 5 means that the results are not reported by the corresponding algorithm.

**Table 5.** Comparison of results in least-waste-first heuristic algorithm (LWF), hybrid simulated annealing algorithm (HSA), iterative bidirectional heuristic placement (IBHP), and hybrid adaptive genetic algorithm (HAGA).

| Instances | LWF | | HSA | | IBHP | | HAGA | |
|---|---|---|---|---|---|---|---|---|
| | F/% | Time/s | F/% | Time/s | F/% | Time/s | F/% | Time/s |
| J1-J2 | - | - | - | - | - | - | 99.96 | 14.45 |
| C11-C73 | 99.4444 | 36.79 | 99.8078 | 41.27 | 99.9273 | - | 99.98 | 232.29 |
| N1-N12 | 99.748 | 29.83 | 99.9113 | 35.23 | 99.97 | - | 99.74 | 913.76 |
| cgcut1-cgcut3 | 95.9151 | 33.42 | 96.5341 | 40.00 | 97.0895 | - | 97.5755 | 99.55 |
| gcut1-gcut13 | 90.6318 | 42.89 | 89.5653 | 60.02 | 89.6397 | - | 89.6145 | 108.36 |
| ngcut1-ngcut12 | 85.5559 | 2.76 | 85.1418 | 25.01 | 85.3714 | - | 86.5549 | 18.38 |
| beng1-beng10 | 99.3778 | 6.07 | 99.3312 | 6.05 | 99.4326 | - | 99.3429 | 228.12 |

From Table 5, it can be observed that the average filling rate of HAGA on the C11-C73, cgcut1-cgcut3, and ngcut1-ngcut12 instances is 99.8%, 97.5755%, and 86.5549%, respectively, which outperforms LWF, HSA, and IBHP. Compared with LWF for these instances, the average filling rate of HAGA can be increased by 0.5356%, 1.6604%, and 0.999%, respectively. The maximum average filling rate can be improved by 1.6604% on cgcut1-cgcut3 for LWF. The average filling rate of HAGA is better than HSA except for N1-N12. Compared with IBHP for C11-C73, cgcut1-cgcut3, and ngcut1-ngcut12 instances, the average filling rate of HAGA can be increased by 0.0527%, 0.486%, and 1.1835%, respectively. Therefore, the HAGA achieved better quality for C11-C73, cgcut1-cgcut3, and ngcut1-ngcut12 instances within reasonable time, with the elimination of equipment and language differences.

However, the average filling rate of HAGA on the N1-N12, gcut1-gcut13, and beng1-beng10 instances is worse than LWF and IBHP. For example, the average filling rate of HAGA is only 0.23%, 0.0252%, and 0.0897% lower on these instances for IBHP, respectively. Overall, the HAGA is relatively superior to these three algorithms within a reasonable duration of computational time, especially on the C11-C73, cgcut1-cgcut3, and ngcut1-ngcut12 instances.

In order to observe the performance of different changes of the proposed algorithm, we just run the proposed algorithm on the N1-N12 instances. We first set $c = 0.1$ and perform the experiments with $\lambda = \{0, 0.3, 0.6, 0.9, 1\}$. For each $\lambda$, the calculation result is the average of twenty operations for each instance. Figure 8 with the $y$-axis representing the average filling rate reports the results of different $\lambda$. From Figure 8, we can observe that $\lambda = 0.6$ is better. Now, we fix $\lambda = 0.6$ and perform the experiments with $c = \{0.05, 0.1, 0.2, 0.4, 0.8, 1.6\}$. Similarly, Figure 9 shows the computational results of different $c$. From Figure 9, we can observe that $c = 0.2$ is better.

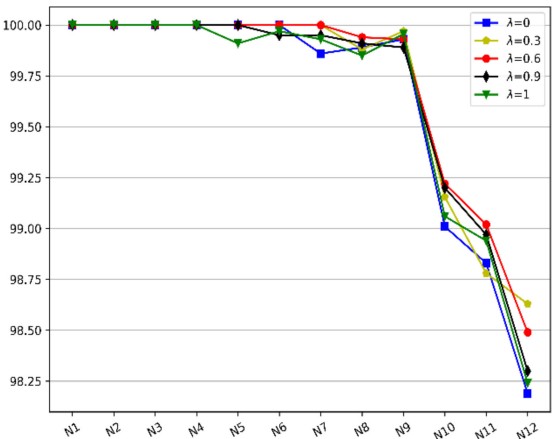

**Figure 8.** The effect of HAGA parameter $\lambda$.

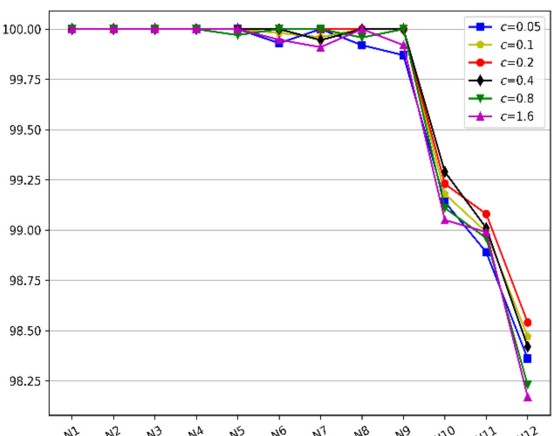

**Figure 9.** The effect of HAGA parameter $c$.

Furthermore, from Figures 8 and 9, it can be seen that the change of parameters has a greater impact on the performance of HAGA as the scale of the instance increases. However, the parameter changes for different problems may achieve better results. In addition, if $\lambda = 0$, the algorithm just changes to AGA [26] from the local perspective of the individual fitness; if $\lambda = 1$, the algorithm happens to be an adaptive genetic algorithm that changes from the overall perspective of the population evolution. We can see that the proposed algorithm, HAGA, has better performance. According to the results of parameter changes, we can establish a consistent parameter setting for HAGA to obtain good benefits.

## 5. Conclusions

In this paper, in order to better solve the 2DRP problem, we analyzed the respective shortcomings of the current two types of adaptive genetic algorithms in detail and proposed an improved method named hybrid adaptive genetic algorithm (HAGA). In the process of providing the solution, the crossover probability and mutation probability of chromosomes are adaptively adjusted according to the collective effect of individual fitness and population evolution process. When using HAGA, the initial population is constructed through a combination of sorting rules and random generation, and operators with better performance are fully used for selection, crossover, and mutation. The heuristic lowest horizontal line algorithm is used as the positioning algorithm to place rectangles. Through continuous iterative optimization of the algorithm, the final packing layout and sheet filling rate are obtained.

The proposed improved method is compared with other representative algorithms. The computational experiments on a wide range of benchmark instances from zero-waste to non-zero-waste problems are implemented. The computational results show that HAGA outperforms the comparison of two adaptive genetic algorithms. Compared with some recent algorithms, the results reveal that HAGA is relatively better than HSA, LWF, and IBHP, especially on the C11-C73, cgcut1-cgcut3, and ngcut1-ngcut12 instances. Moreover, in view of observing the performance of different changes, the experiments were designed a series of $\lambda$ and $c$ parameters to determine the consistent parameter settings for HAGA. The presented algorithm, which works adaptively through the two perspectives of individual and population evolution, can also be applied to cutting, the Travelling Salesman Problem (TSP), Vehicle Routing Problem (VRP), etc. In the future, the focus of research is to further improve the calculation speed and filling rate of the algorithm for 2DRP problem and strive to find a satisfactory solution for larger scale and difficult benchmark instances.

**Author Contributions:** Conceptualization, Y.-B.L. and X.X.; methodology, H.-B.S. and Y.-B.L.; software, H.-B.S. and X.X.; validation, H.-B.S. and Y.-B.L.; data curation, Y.-R.L. and X.X.; writing—original draft, Y.-R.L. and H.-B.S.; writing—review and editing, H.-B.S. and Y.-B.L.; supervision, Y.-B.L. and X.X. All authors have read and agreed to the published version of the manuscript.

**Funding:** This work was supported by National Key Research and Development Program of China (No.2018YFF0212201).

**Institutional Review Board Statement:** Not applicable.

**Informed Consent Statement:** Not applicable.

**Data Availability Statement:** The test source data comes from an existing data set. The test source data can be downloaded from http://people.brnel.ac.uk/~mastjjb/jeb/info.html or related literature. The test source data does not belong to us. The experimental result data is calculated by applying the algorithm of this paper to the test source data.

**Conflicts of Interest:** The authors declare no conflict of interest.

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
