# Peer review of "An Improved Adaptive Genetic Algorithm for Two-Dimensional Rectangular Packing Problem"

_applsci, doi:10.3390/app11010413_

Round 1

Reviewer 1 Report

The manuscript studies the two-dimensional rectangular packing problem. After introducing the topic and reviewing the main literature for their purposes, the authors formulate an adaptive genetic algorithm for solving this problem. Through extended computational experiments led on a number of instances from the literature, the authors pretend to show that their algorithm outperforms the ones from the literature.

The topic of the manuscript is suitable for the journal and it is interesting. However, multiple concerns arise and some of them do not allow to express a clear opinion on the manuscript. The most critical point concerns the results and their presentation. The authors have not scaled the results from the papers from the literature and, consequently, it is hard to say if their methodology really outperforms the approaches from the literature. Moreover, the manuscript should be revised from the language point of view. Specific concerns are listed below. My recommendation is for a major revision.

  • I would not put examples already in the introduction to explain the methodology
  • The citation style is not consistent: sometimes the authors use the name of the authors of the papers, other times only numbers.
  • Double check the language correctness (for example, at page 11: “The all algorithms” should be “All the algorithms”, at page 13 the use of singular and plural forms in the sentence “The scale and difficulty…filling rate of 100%”). Furthermore, there are some repetitions.
  • In section 4, the authors use inappropriate terms:
    • If they state that the number of iterations has been set to 2000, it is not necessary to add the word “times”.
    • Moreover, the calculation results are the average of twenty “runs”, not “operations”.
  • The authors should double check the layout of the text (some words are not aligned with the rest, see, for example, the rectangle dimension at the end of page 11).
  • Concerning Tables 4, 5, and 6 the authors state that their algorithm is faster with respect to the ones from the literature. However, multiple concerns arise. First, it has been shown that the programming language used has an impact on runtimes. From the manuscript, it is not clear which is the programming language used for coding the algorithms from the literature. Second, it is not clear if the time reported in Tables 4, 5, and 6 has already been scaled in order to compare with authors’ algorithm performance and make the results comparable in terms of resources (processors, RAM) used. I suggest them to report the scalarized times in Tables 4, 5, and 6 to make the comparison valid. They can have a look at the following website: https://www.passmark.com/. Furthermore, algorithm HAGA is not always dominating the variants from the literature (see, for example, the results for instance group J2, in which the solution quality provided by algorithm AGA is better than HAGA, or instance group C62 in Table 5).
  • At the end, the authors state that their algorithm could be used also for vehicle routing problems. Because of this statement, I invite the authors to carry on a more detailed literature view to compare their algorithm with the ones developed in the literature for different classes of problems (for example, see “A hybrid genetic algorithm with adaptive diversity management for a large class of vehicle routing problems with time-windows. T Vidal, TG Crainic, M Gendreau, C Prins - Computers & operations research, 2013”).

Reviewer 2 Report

Congratulations to authors. This is very good paper.

When trying to help authors in their efforts to further improve the paper, only such a proposal can be recommended: the obtained results should be discussed a little more precisely and compared with other similar studies.

Good luck.

Author Response

Thank you very much for your approval. Based on your valuable comments, we make efforts to further improve the paper. We discussed a little more precisely and compared with other similar studies in 4.3. Compare the results with other algorithms.